# Personal Network Inference Unveils Heterogeneous Immune Response Patterns to Viral Infection in Children with Acute Wheezing

**DOI:** 10.3390/jpm11121293

**Published:** 2021-12-03

**Authors:** Laura A. Coleman, Siew-Kim Khoo, Kimberley Franks, Franciska Prastanti, Peter Le Souëf, Yuliya V. Karpievitch, Ingrid A. Laing, Anthony Bosco

**Affiliations:** 1Medical School (Paediatrics), University of Western Australia, Perth, WA 6009, Australia; laura.coleman@uwa.edu.au (L.A.C.); peter.lesouef@uwa.edu.au (P.L.S.); Ingrid.Laing@telethonkids.org.au (I.A.L.); 2Telethon Kids Institute, University of Western Australia, Perth, WA 6009, Australia; Kim.Khoo@telethonkids.org.au (S.-K.K.); Kimberley.Franks@outlook.com (K.F.); Franciska.Prastanti@telethonkids.org.au (F.P.); Yuliya.Karpievitch@telethonkids.org.au (Y.V.K.); 3School of Biomedical Sciences, University of Western Australia, Perth, WA 6009, Australia

**Keywords:** acute asthma, network inference, transcriptome

## Abstract

Human rhinovirus (RV)-induced exacerbations of asthma and wheeze are a major cause of emergency room presentations and hospital admissions among children. Previous studies have shown that immune response patterns during these exacerbations are heterogeneous and are characterized by the presence or absence of robust interferon responses. Molecular phenotypes of asthma are usually identified by cluster analysis of gene expression levels. This approach however is limited, since genes do not exist in isolation, but rather work together in networks. Here, we employed personal network inference to characterize exacerbation response patterns and unveil molecular phenotypes based on variations in network structure. We found that personal gene network patterns were dominated by two major network structures, consisting of interferon-response versus FCER1G-associated networks. Cluster analysis of these structures divided children into subgroups, differing in the prevalence of atopy but not RV species. These network structures were also observed in an independent cohort of children with virus-induced asthma exacerbations sampled over a time course, where we showed that the FCER1G-associated networks were mainly observed at late time points (days four–six) during the acute illness. The ratio of interferon- and FCER1G-associated gene network responses was able to predict recurrence, with low interferon being associated with increased risk of readmission. These findings demonstrate the applicability of personal network inference for biomarker discovery and therapeutic target identification in the context of acute asthma which focuses on variations in network structure.

## 1. Introduction

Acute respiratory viral infections are a leading cause of wheezing and asthma exacerbations in children, with a virus often identified in more than 80% of cases [1,2,3]. Rhinovirus (RV) is the most common infectious agent identified during an acute respiratory illness and particularly in acute childhood asthma, having been detected in up to 87.5% of cases [1,2,3,4,5]. RV is a single-stranded RNA virus, a member of the *Picornaviridae* family [6,7]. There are three species of RV (A, B and C) [8]. RV-C is frequently the most prevalent species in children presenting to hospital with acute asthma [2,9], especially in preschool children [10], although RV-A is the most prevalent species in milder viral wheezing in children [11,12]. RV-C has been associated with a higher severity of exacerbations compared to RV-A or RV-B [2,13]. RV-C enters host cells via the receptor cadherin 3 (CDHR3), unlike RV-A, which uses either intercellular adhesion molecule 1 (ICAM-1) or low-density lipoprotein receptor (LDLR), and RV-B, which uses ICAM-1 [14]. Genetic variants in *CDHR3* leading to an increased cell surface expression have been associated with recurrent severe asthma exacerbations in children [15,16]. The increased pathogenesis of RV-C is likely due to multiple factors, including the altered structure of the viral capsid compared to RV-A and RV-B that appear to make RV-C less detectible by the host immune system and anti-viral drugs [17,18,19], and the lesser ability of the host immune system to make antibodies specifically against RV-C [20,21]. However, the mechanisms by which RV-C triggers increased exacerbation severity and recurrence remain to be fully elucidated. RV-A can also trigger wheezing exacerbations, although with a similar or lower prevalence compared to people with common cold in the community [22,23,24,25,26,27]. RV-B infections are less frequent and less pathogenic, due to slower replication and reduced cytokine and chemokine induction [28].

Previous gene expression profiling studies of childhood asthma exacerbations have focused on identification of differentially expressed genes between samples collected from children during the acute illness compared with convalescence, or alternatively for more severe cases compared with less severe cases [29,30,31,32]. Moreover, employing cluster analysis of gene expression profiles we demonstrated that immune response patterns during acute exacerbations of asthma/wheeze were highly heterogeneous and could be divided into IRF7-high versus IRF7-low molecular phenotypes [33]. Notably, these previous studies focused on differential gene expression, and this approach is limited, since genes do not exist or function in isolation (they work together in networks). Accordingly, there is a growing understanding in the systems biology literature that the mechanisms that determine health versus disease are more related to changes in network structure as opposed to variations in gene expression levels [34].

In transcriptomics studies, network structure is typically measured by calculating gene co-expression patterns across a large cohort of samples. This approach results in the identification of an aggregate network that is essentially averaged over a large number of subjects. A limitation of building aggregate networks is that you cannot capture patterns in the data that are restricted to a small subgroup of patients or an individual subject. This limitation is addressed by personal network inference algorithms that can model gene network patterns at single subject resolution. Personal network inference considers the group aggregate network as the linear combination of the individual networks of each group member, and so by taking the difference between the aggregate network and the aggregate minus one individual, a personal network can be constructed [35]. Here, we employed personal network inference to unveil heterogeneous molecular phenotypes underlying RV-C or RV-A induced wheezing in children. Our findings reveal a unique level of insight about the mechanisms underlying acute asthma that are not detectable employing conventional differential expression analyses.

## 2. Materials and Methods

Children aged 0–18 years were recruited into the Mechanisms of Acute Viral Respiratory Infection in Children (MAVRIC) cohort upon presentation to the emergency department of Princess Margaret Hospital (PMH) with an acute lower respiratory tract illness (ALRI). This study was granted ethics approval from PMH (Ethics #1761EP) and informed consent was obtained from at least one parent/guardian prior to recruitment. Controls were recruited from siblings or friends of cases, and from the community. They were not required to be free of respiratory pathogens or upper respiratory symptoms.

At recruitment, children completed a skin prick test, and blood samples and nasal swabs were collected. Asthma exacerbation severity score, atopy, and total and specific IgE were assessed as described previously [33]. The presence of respiratory viruses and specific pathogenic bacteria were assessed using a tandem multiplex RT-PCR assay [36]. RV species were identified by genotyping a 270-bp variable sequence in the 5′ non-coding region of the RV genome, and RV species and genotypes were assigned as previously described [37]. Clinical characteristics were compared using t tests or two-proportion z tests as appropriate.

Nasal epithelial samples were collected using flocked swabs (Copan Diagnostics Inc., Murrieta, CA, USA). Swabs were stored in TRIzol (Invitrogen, Carlsbad, CA, USA) at −80 °C, and total RNA was extracted using RNeasy (Qiagen, Hilden, Germany). RNA quantity and quality were assessed using a Bioanalyzer RNA 6000 Nano Chip (Agilent, Santa Clara, CA, USA) and NanoDrop 2000 spectrophotometer (Thermofisher Scientific, Waltham, MA, USA) [33]. Of the children recruited with nasal samples available, 120 contained enough RNA of sufficient quality for microarray. The RNA was processed and hybridised to genome-wide gene expression micro-arrays (Human Gene ST2.1; Affymetrix, Santa Clara, CA, USA). The micro-array hybridisation was performed by The Ramaciotti Centre for Genomics at the University of New South Wales (GSE103166).

Microarray data on nasal samples were analysed in the open-source statistical software R (www.r-project.org/ accessed on 28 July 2020). The microarray data were pre-processed employing the RMA algorithm, using custom mapping of microarray probe-sets to the genome (hugene21sthsentrezgcdf Version 19) [38]. The quality of the microarray data was assessed using the R package arrayQualityMetrics, and nine low quality samples were removed from the analysis. Differentially expressed genes were identified employing the limma (LInear Models for Micro-Array) R package, with false discovery rate (FDR) control for multiple testing using Benjamini Hochberg correction [39,40,41]. The linear models were adjusted for batch effects and other hidden confounders with sva (surrogate variable analysis) [42]. The sva package estimates variation in expression data caused by biological and technical factors that may not be measurable, allowing them to be included in the linear models used by limma [42,43]. The probe-sets were filtered with the pvac package, which calculates the proportion of variance of a probeset explained by the first principle component, with a cut-off value of 0.5 [44]. The pvac algorithm assesses the consistency of expression levels across probes measuring the same gene and then removes genes from analysis if they are inconsistent. The comparisons made were RV-A-infected, RV-C-infected and RV-negative cases, restricted to only cases with wheeze as noted by the attending physician, compared with RV-negative controls and RV-negative convalescence samples. The significant genes from any of the comparisons were combined to form a panel of 646 exacerbation signature genes.

Personalised networks were constructed for each individual subject by applying Linear Interpolation to Obtain Network Estimates for Single Samples (LIONESS) using the lionessR package [45]. Expression data were first corrected for technical variation using the RUVcorr and RUVnormalize packages, with smoothing set at 0.2, k at 20 and nu coefficient at 0, and 100 negative control genes selected [46,47]. The genes previously identified as differentially expressed from the limma analysis were selected for personalised network analysis. Individual gene co-expression networks were extracted using lionessR. The top one percent of network edges by edge weight were plotted using the software Gephi [48]. Genes were grouped into pathways using upstream regulator analysis in the Ingenuity Pathways Analysis software (Qiagen Digital Insights, Aarhus, Denmark). IRF-high/low classification was made by examining the proportion of network edges assigned to interferon signalling pathways. Functional pathways were also examined using InnateDB pathway over-representation analysis [49].

Differences in proportions of network edges between groups were compared using non-parametric t tests. The network proportions were also used in a principal components analysis to cluster the acute cases using FactoMineR [50]. The top 20 most connected genes (“hubs”) in each network were used to construct word clouds for each cluster. Clinical characteristics were compared using t tests or two-proportion z tests as appropriate. Survival analysis of time until next hospitalisation was conducted using the survival [51] and survminer [52] R packages. Rank plots of eigengenes against network proportions were used to test for relationships between gene expression and network topology.

We then went on to investigate the interferon high/low gene signature in a second cohort (GSE115824). RNA-seq data were processed as per the original paper, using voom to normalise the RNA-seq count data whilst taking into account and adjusting for the mean-variance relationship of each gene [53]. Individual networks were constructed and pathways classified as before using the signature genes present in the data (544 genes present/646 total).

## 3. Results

We stratified the population into five groups, comprising RV-C cases, RV-A cases, RV-negative cases, RV-negative convalescence and RV-negative controls. The clinical characteristics of the study population are shown in Table 1. As per the inclusion criteria, controls and subjects sampled during convalescence had not been diagnosed with acute asthma or wheeze. However, one or more viruses was detected in 38% of the controls and 21% of the convalescence samples.

Other characteristics were similar except that the RV-C cases subset were younger compared with the RV-negative controls (RV-C: 4.11 years (SD 3.09), RV-negative controls: 6.62 years (SD 4.50), *p <* 0.05). The RV-negative cases were also younger than the RV-negative controls (RVneg: 3.47 years (SD 3.26), *p <* 0.05), as well as the RV-A-infected cases (RV-A: 6.42 years (SD 3.30), *p <* 0.05). The proportion of children with atopy was higher in the RV-A-infected cases compared with the RV-negative cases (RV-A: 91%, RVneg: 42%, *p <* 0.01), as was the proportion of children with atopy to aeroallergens (RV-A: 82%, RVneg: 42%, *p <* 0.05) (Table 1).

Gene expression profiles from nasal epithelial samples were available from all subjects presented in Table 1. We initially analyzed the data employing conventional statistical methods to identify differentially expressed genes between all experimental groups. We identified a total of 646 differentially expressed genes, which were selected for further analysis (Appendix A). The signature included genes associated with interferon responses, innate immunity, and type 2 inflammation (e.g., *IRF7*, *IFIT1-3*, *IFNG*, *TLR4*, *TLR8,* and *IL33*). A subset of the genes have been validated using qRT-PCR, specifically RSAD2, MX1, DDX60, IRF7, ISG15, THBS1, CD163, IL18R1. TLR2, FCER1G, ARG1, IL1R2, IL33, and NCR1 [33]. We performed hierarchical cluster analysis on this signature to examine the global patterns in the data, and the results revealed a clear separation of cases and controls as expected (Figure 1).

We then analyzed the gene signature using *LIONESS*, to extract personal gene network patterns from the data. We retained the top 1% of network edges for each individual subject and summarized the proportion of network edges from each personal network at the signaling pathway level using Upstream Regulator Analysis. Notably, these pathways represent known signaling cascades that are controlled by defined molecular drivers. The dominant pathways identified in the gene signature were core interferon, type I interferon and dexamethasone. Cluster analysis of these data employing FactoMineR revealed that there were four distinct clusters in the data (Figure 2). We summarized the proportion of network edges assigned to each pathway as pie charts, and hubs were summarized across subjects within each cluster as word clouds. The data showed that the personal gene network pathway signatures and hubs differed between groups. For example, prominent hubs for cluster A were *EPSTI1*, *IFIH1*, *EIF2AK2*, *IFIT3*, *CMPK2*, *IFIT2*, and *DDX58*. Cluster B was enriched with the hubs *DDX58*, *IFIT1*, *MX1*, *EIF2AK2*, *IFIT5*, and *IFIT3*. Cluster C was the most distinct group and was marked by the hubs *FCER1G*, *MCEMP1*, and *DYSF* and greatly reduced representation of interferon-associated genes. The most prominent hubs for Cluster D were *IFI44*, *IFIT1*, *IFI44L*, *EIF2AK2*, *ISG15*, *OAS3*, and *DDX60* (Figure 2).

Cluster A had both the highest proportion of type 1 and 3 interferon-related pathways and highest combined proportions of all interferon-associated pathways. Cluster B had lower proportions for interferon pathways than Cluster A, followed by Cluster C and finally Cluster D that had the lowest proportions (Figure 2). Notably, Cluster C had the highest proportion of dexamethasone pathway genes, but generally, a decrease in interferon pathway genes corresponded with an increase in genes for the category “other” which was not enriched for any known biological pathways. We employed a series of bioinformatics tools to further probe the biological function of the genes from the “other” pathway category. This analysis revealed that the “other” pathway category was associated with the synthesis and metabolism of membrane lipids and arachidonic acid, and phagocytosis. 

We then examined the clinical features of children from the four clusters. The data showed the clusters differed in the prevalence of atopy, which was lower in Cluster D compared with Cluster A (36% vs. 74%), as well as aeroallergen sensitization, which was lower in Cluster D compared with Clusters A and B (27% vs. 68% and 69%) (Table 2). There was no difference between the groups in relation to the detection of RV-C or RV-A.

We also examined recurrent illness. Survival analysis of time until next hospitalization showed that although visually, Cluster C appeared to have the shortest time until a second wheezing episode, and being approximately twice as likely to experience a second episode than Clusters A and D, this was not significantly different (*p* = 0.55); most likely this was due to the low numbers within each cluster (Figure 3A). The difference in likelihood of and time to readmission was most pronounced when Cluster A and D combined were compared with Cluster C (Figure 3B). When the survival curve was split by dividing children into quartiles based on the ratio of interferon-associated edges to *FCER1G*-associated edges, we found that children with the lowest IFN:*FCER1G* were more likely to have a second admission and had a shorter time until the next admission (Figure 3C). These differences in the incidence of and time until readmission were most obvious when the upper three quartiles were combined: members of the first quartile were twice as likely to experience a second admission (*p* = 0.018) (Figure 3D).

We then examined variations in personal network structure within each cluster. For cluster A, gene network patterns in a subset of the children were completely dominated by interferon-associated genes (Figure 4A), whereas in other children the patterns consisted primarily of interferon-associated genes alongside a second, less prominent module featuring *FCER1G*, *DYSF*, *CST7*, and *MCEMP1* (Figure 4B).

In Cluster B, children had networks consisting primarily of interferon-associated genes, but with greater representation of the second *FCER1G* module seen in Cluster A. For some children, these gene modules overlapped (Figure 5A), whereas in others they remained isolated (Figure 5B).

In Cluster C, personal gene networks were dominated by the *FCER1G*-associated module, but a module of interferon-related genes was also present. As in Cluster B, these modules were overlapping in some children (Figure 6A) but isolated in others (Figure 6B).

Finally, in Cluster D, networks were highly heterogeneous, comprising an interferon-associated gene module, and often a second *FCER1G* module and a third module, the composition of which was highly variable (Figure 7A,B).

It has been demonstrated that changes to network structure can occur in the absence of differential gene expression [34]. To investigate this issue in the current study, we superimposed the cluster membership derived from personal gene network patterns over the heatmap from Figure 1. The data showed that whilst a subset of subjects from Cluster 1 were co-localized in the heatmap, the bulk of the subjects were randomly distributed throughout the dendrogram (Figure 8). Moreover, we calculated ranks for pathway-level data based on gene expression (eigengenes) or personal network pathway signatures, and the data were independent (Appendix A).

We also examined personal gene network patterns underlying acute wheezing and their dynamic states in a data set from public domain. The data contained nasal transcriptomes from children with acute viral wheeze sampled at enrolment (with no exacerbation), and on days one to three and four to six during the acute illness. The characteristics of the study population have been previously described [54]. We applied personal network inference to interrogate gene network patterns in children experiencing a virus-induced exacerbation or controls, stratified by time point using our gene signature of 646 genes. At baseline, two clusters/groups were identified (Figure 9). The hub genes and interferon pathway proportions varied between the two groups, with Cluster A having *PARP14*, *IFIH1*, *DDX58* and *IFIT2* as major hubs and a higher proportion of both interferon types 1 and 3 genes and total interferon-related genes compared with Cluster B, which had *PERP* as a prominent hub. Of note, the dexamethasone and “other” pathways were increased in Cluster B, similar to what we observed in the MAVRIC cohort where the proportion of interferon pathways was decreased. 

During acute illness (day one to three), two clusters were identified (Figure 10). The hub genes and interferon pathway proportions varied between the two groups, with Cluster A having *STAT2*, *ZBP1* and *DHX58* as major hubs and a higher proportion of both interferon types 1 and 3 genes and total interferon-related genes compared with Cluster B, which had *TAP1* as a prominent hub. As before, we observed a shift between interferon pathway genes which were decreased and dexamethasone and “other” pathways which were increased.

For the late visit (Day 4–6), three clusters were identified (Figure 11). The hub genes and interferon pathway proportions varied between the three groups, with Cluster A having *STAT2*, *ZBP1*, *OASL* and *PML* as major hubs, and a higher proportion of both interferon types 1 and 3 genes and total interferon-related genes compared with Cluster B, which had *IFIT3*, *GNS*, *RALB*, *FCER1G*, and *DDX58* as prominent hubs, and Cluster C, which had *SERPINA1*, *IFIH1*, and *AQP3* as major hubs. Decreases in the interferon pathways once again corresponded with increases in the dexamethasone and “other” pathways.

## 4. Discussion

Rhinovirus-induced exacerbations of asthma and wheeze are a major cause of emergency room presentations and hospital admissions among children. Previous studies have demonstrated that immune response patterns during these illnesses are highly heterogeneous and are characterised by the presence or absence of robust interferon responses. Here, for the first time we employed personal network inference to unveil heterogeneous response patterns on the basis of variations in network structure at single subject resolution. Our findings demonstrate that personal gene network patterns were dominated by two major network structures: comprising interferon-response versus *FCER1G*-associated networks. Moreover, cluster analysis of these structures divided the children into subgroups, which differed on the basis of atopy but not RV species. We also demonstrated that the network structures could be observed in an independent cohort of children with virus-induced asthma exacerbations that were sampled over a time course, and we showed that the *FCER1G*-associated networks were mainly observed at late time points (days four to six) during the acute illness. Finally, we showed that the ratio of interferon- and *FCER1G*-associated gene network responses was able to predict the likelihood of recurrence, where children with low interferon were most likely to be readmitted to hospital for wheeze. In summary, employing personal network inference, we have identified two major gene network patterns that stratify the population into molecular phenotypes. Our findings suggest that next generation biomarker discovery and drug development programs for acute asthma/wheeze should focus on variations in network structure rather than differential gene expression.

Reconstruction of the network wiring diagrams from the MAVRIC cohort indicated that the four subgroups were defined based on the balance between the interferon-associated module and a second module associated with *FCER1G*, *DYSF*, *MCEMP1*, and *CST7*. The networks of the first group were dominated almost entirely by interferon-related genes, those of the second group were weighted towards interferon-associated genes but had greater representation of the second module, and the third group’ networks were weighted in favour of non-interferon genes. The group where the host response is skewed away from interferon-related genes towards other pathways is at increased risk of recurrence, particularly those with a low ratio of interferon- to *FCER1G*-associated edges. The fourth group did not fit this pattern, having low representation of interferon-associated genes, but not skewing strongly towards the other module. Instead, the networks in this group were much more heterogeneous and unfocused. This observation highlights a unique characteristic of *LIONESS*/personal network inference, which can extract biological insights in the presence of highly heterogeneous responses.

Our analysis of the data from the external validation cohort published by Altman and co-workers, showed that subgroups were also defined by the balance between interferon-associated genes and other modules. At the baseline and acute visits, the subgroups were interferon-high and -low. In comparison, the late visit had one interferon-high group and two interferon-low groups that were centred around different hub genes. The first interferon-low group contained several hub genes also observed in the *FCER1G/DYSF* gene module, indicating this group corresponds to cluster C from the previous analysis, whereas the hub genes of the second interferon-low group were more heterogeneous, as observed for cluster D from the MAVRIC cohort. The observation that the *FCER1G/DYSF* module was prominent at the late visit is consistent with the recruitment of *FCER1G*-bearing myeloid cell populations from the circulation [32].

This study expands previous work, where unsupervised cluster analysis was used to identify subgroups of children with acute wheeze [33]. The key gene module that divided acute cases into two groups was reconstructed employing prior knowledge and was centred on *IRF7*, a key regulator of interferon expression in response to viral infection. *IRF7* was also a major component of the core interferon pathway described in this study, where network construction was based on a data-driven approach. Whilst these findings demonstrate that a variety of methods can be employed to identify interferon-high and –low response phenotypes during exacerbations, the *LIONESS* approach is the only method that can extract gene network patterns at single subject resolution.

The second most prominent gene module was enriched with biological functions associated with the activation of platelets, neutrophils, leukocytes, eosinophils and mast cells (including *FCER1G*, *MCEMP1*, *LILRA5*, *CR1*, *SELL*, *FPR2*, *CD53*, *CLEC4D*, *CLEC4E*, *ALOX5*, *IL17RA*, and *SERPINA1*) [55]. Children from MAVRIC cluster D had the lowest representation of this module and also the lowest prevalence of atopy, suggesting that this module is associated with allergic inflammation. Similarly, throughout both cohorts and across multiple time points, there was a persistent trend where decreases in the proportion of interferon-associated genes within networks corresponded to an increase in the dexamethasone pathway, featuring genes such as *ADAM9*, *AQP5*, *CD177*, *CD48*, *CR1*, *IL33*,and *NLRC4*, and the “other” pathway. This group of “other” pathway genes was enriched with biological pathways associated with the synthesis and metabolism of membrane lipids and arachidonic acid, suggesting a role in inflammation through lipid mediators such as prostaglandins and leukotrienes.

Our findings suggest that when investigating immune responses to respiratory viral infection and acute wheeze and asthma in children, personalised methods are required in order to capture both group-wise and individual disease mechanisms. We did not, however, find any differences between children with specific virus species, such as RV-A and RV-C. It is not surprising that we could not tease these apart, given that there are multiple immune response phenotypes producing opposing patterns of gene expression, resulting in an overall lack of differential expression between virus species groups. Due to the almost equal prevalence of children with interferon-high and -low phenotypes in both virus groups, sample sizes were too small for meaningful comparison. Follow-up studies with increased numbers of subjects will be required to determine if there are distinct gene network patterns associated with RV-C versus RV-A exacerbations.

Personal network inference could potentially be transformative for biomarker discovery and drug development for highly heterogeneous diseases such as asthma. For example, a previous trial of omalizumab to prevent asthma exacerbations in children found that the treatment had the greatest benefit in children sensitized to cockroach or house dust mite. Omalizumab treatment was able to markedly reduce seasonal exacerbations, but not by preventing viral infections, which were equally prevalent in the treatment and placebo groups [56]. Personal network inference would enable the identification of network structures associated with treatment effectiveness (e.g., *FCER1G*), providing a powerful tool to stratify patients, monitor response to therapy and conduct mechanism of action studies.

Developments in drug discovery techniques using network pharmacology have led to the concept of network-based polypharmacy, where multiple nodes are selected from a network and one or more drugs targeting these nodes are administered, leading to more widespread perturbation of the system through the disruption of multiple hubs at once [57]. These new techniques are exciting given their potential to be combined with personalised network analysis, creating a new paradigm of personalised medicine, where therapies can be generated in response to an individual disease pattern. These new therapies would be able to target and alter the behaviour of entire networks and functional pathways, hopefully leading to greater efficacy of treatment.

Our study has several limitations that we acknowledge. The study participants in the MAVRIC cohort are recruited upon presentation to emergency departments with acute illness, and therefore it is not possible to control for variations in the timing between the onset of infection and the expression of severe symptoms. A subset of the controls was related to the cases, which can potentially lead to biased estimates of variability and effect sizes in the data. The sample size of the study was limited, which makes it difficult to identity relationships between clinical traits and molecular phenotypes. Another limitation was that the gene expression profiling was performed using microarray technology on a mixed population of cells derived from nasal swab specimens. Future studies should employ bulk-RNA-Seq and single cell RNA-Seq to improve the sensitivity, accuracy, and resolution of these analyses. Notwithstanding these limitations, we have used personalised network inference to identify distinct immune phenotypes in children with acute wheeze following respiratory viral infection, as well as predictors of recurrence. Future work will go on to further characterise these immune phenotypes, with the ultimate goal of developing new tools for the identification of children at increased risk of severe or persistent respiratory disease and early intervention.

## Figures and Tables

**Figure 1 jpm-11-01293-f001:**
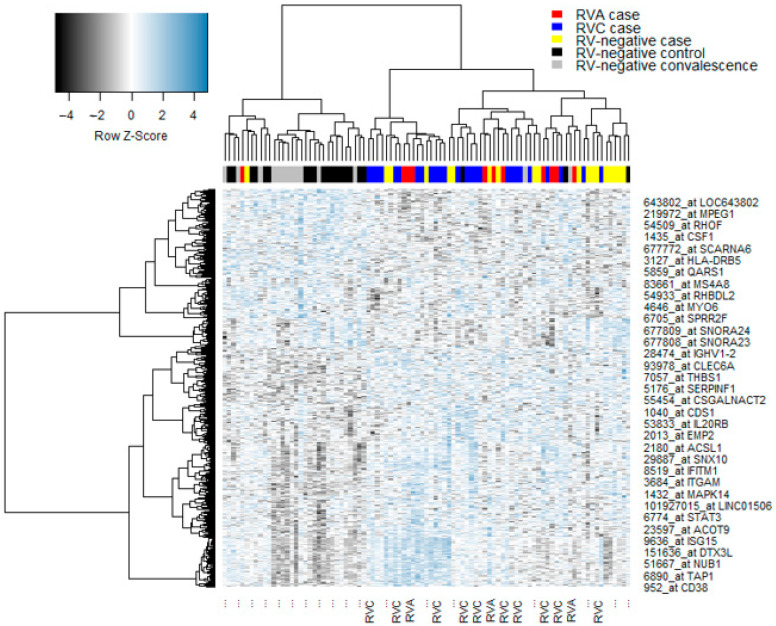
Gene expression was profiled in nasal swab samples from children with acute asthma/wheeze or controls. Differentially expressed genes were identified in children with acute wheezing, convalescence or controls, stratified by RV infection status. The data were transformed to Z-scores, analyzed by hierarchical clustering and visualized as a heatmap.

**Figure 2 jpm-11-01293-f002:**
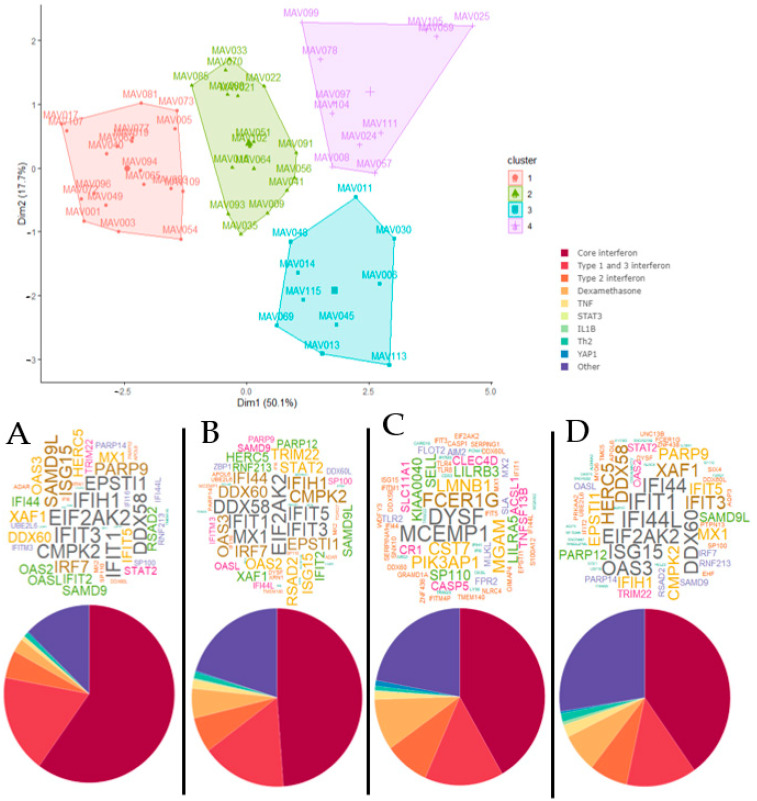
Cluster analysis of personal gene network patterns unveils molecular subtypes of acute wheezing. FactoMineR cluster plot, word cloud plots of the top 20 hub genes for each individual case (**top**) and averaged personal gene network pathway proportions (**bottom**) for clusters A, B, C and D. Word color and size indicates frequency.

**Figure 3 jpm-11-01293-f003:**
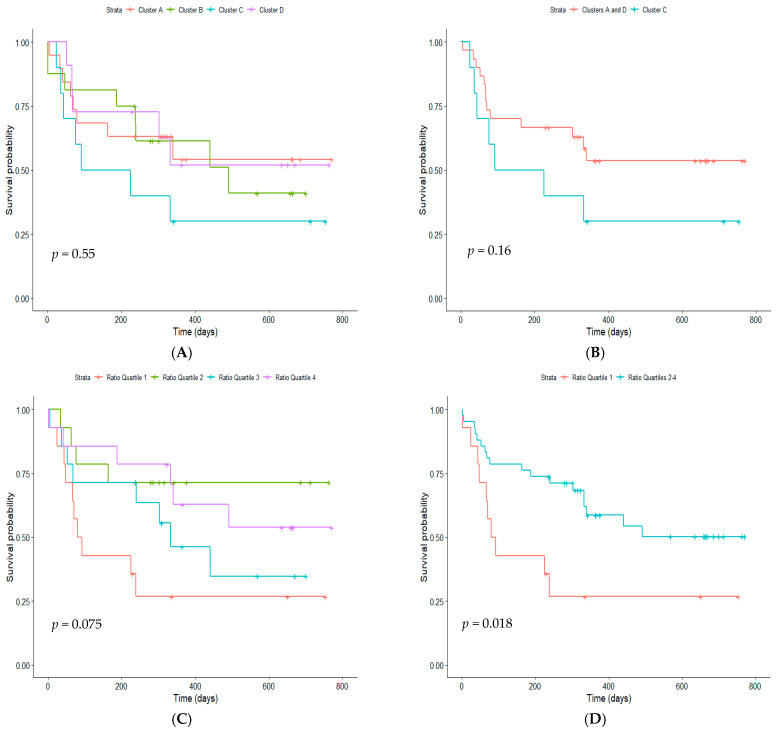
Survival curves of time (days) until next hospitalization for (**A**) Clusters A, B, C and D, (**B**) Clusters A and D combined and Cluster C, (**C**) IFN:*FCER1G* quartiles 1–4 and (**D**) IFN:*FCER1G* quartile 1 and quartiles 2–4 combined.

**Figure 4 jpm-11-01293-f004:**
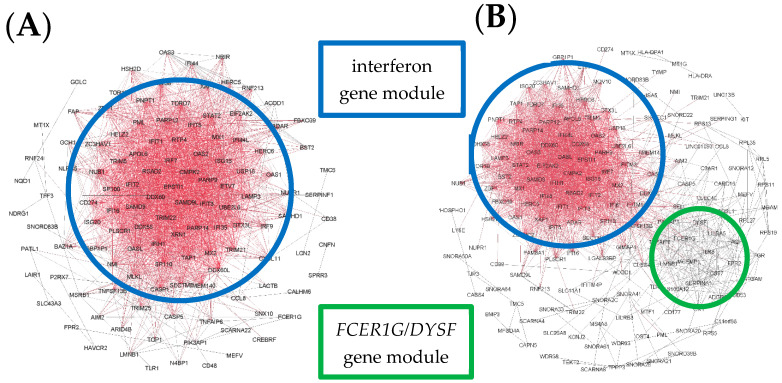
Illustrative examples of personal gene network structures generated in Gephi for representative subjects from Cluster A. Example (**A**) shows an interferon-dominated network, whereas Example (**B**) shows a network with two prominent modules. Red indicates highly connected genes (“hubs”) and black denotes genes with few connections.

**Figure 5 jpm-11-01293-f005:**
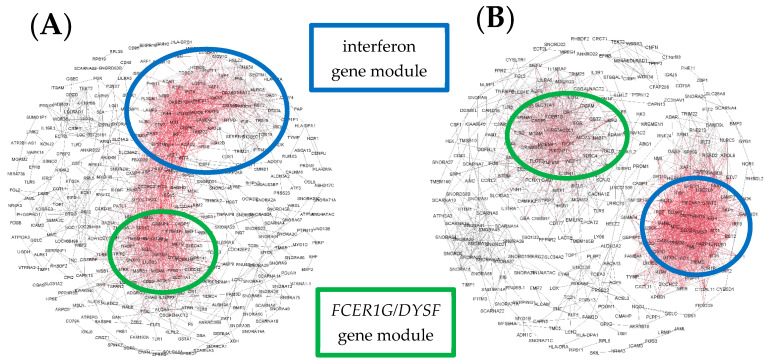
Illustrative examples of personal gene network structures generated in Gephi for representative subjects from Cluster B. Example (**A**) shows a network with overlapping modules, whereas Example (**B**) shows a network with two isolated modules. Red indicates highly connected genes (“hubs”) and black denotes genes with few connections.

**Figure 6 jpm-11-01293-f006:**
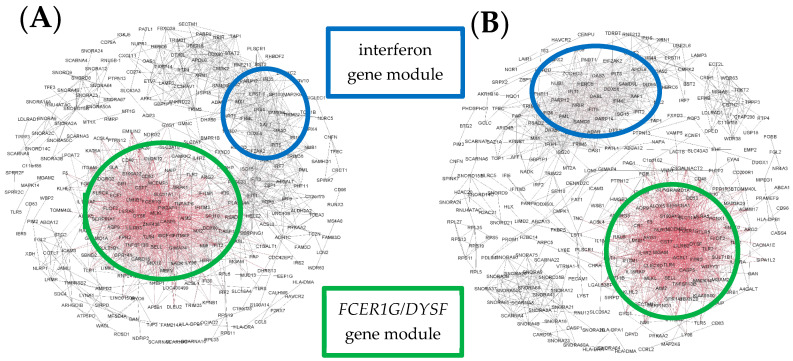
Illustrative examples of personal gene network structures generated in Gephi for representative subjects from Cluster C. Example (**A**) shows a network where the modules overlap, whereas in Example (**B**) the modules are isolated. Red indicates highly connected genes (“hubs”) and black denotes genes with few connections.

**Figure 7 jpm-11-01293-f007:**
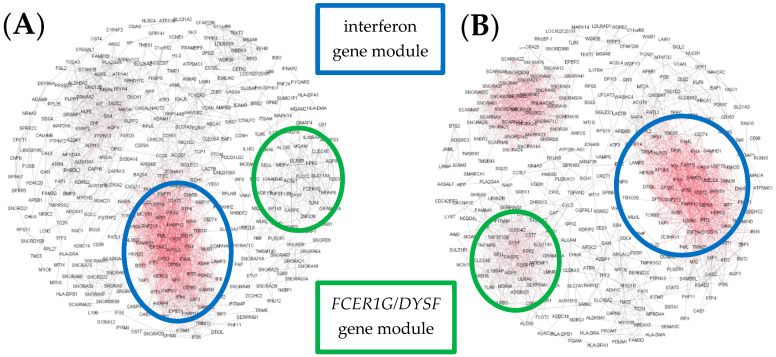
Illustrative examples of personal gene network structures generated in Gephi for representative subjects from Cluster D. Examples (**A**,**B**) show networks with more than two prominent modules. Red indicates highly connected genes (“hubs”) and black denotes genes with few connections.

**Figure 8 jpm-11-01293-f008:**
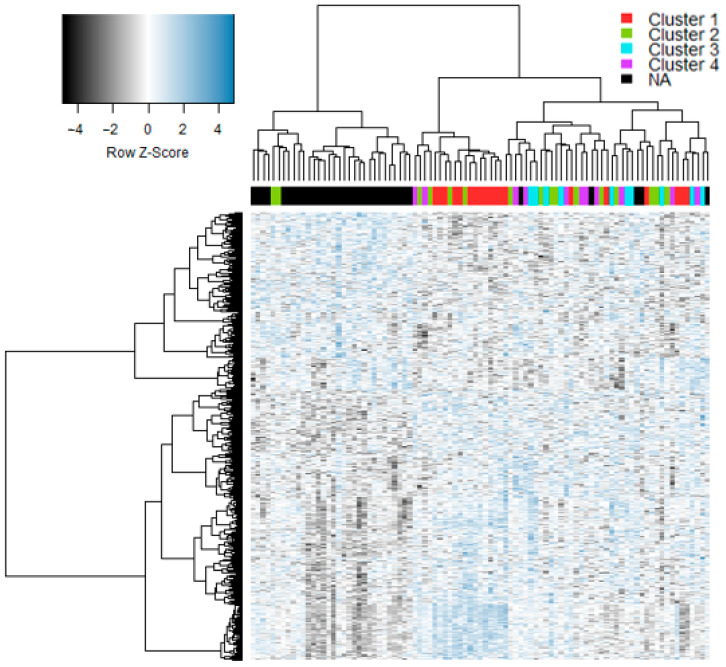
Heatmap of gene signature in case, convalescence and control samples, colored by cluster membership. NA = not applicable (control or convalescence sample).

**Figure 9 jpm-11-01293-f009:**
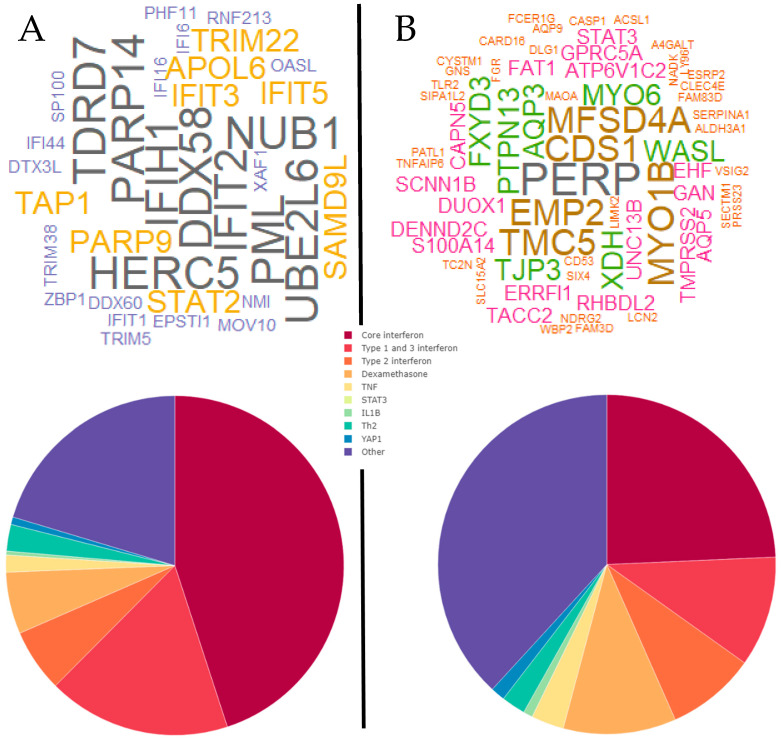
Word cloud plots of the top 20 hub genes for each individual at the baseline visit (**top**) and averaged network pathway proportions (**bottom**) for clusters A and B. Word color and size indicates frequency.

**Figure 10 jpm-11-01293-f010:**
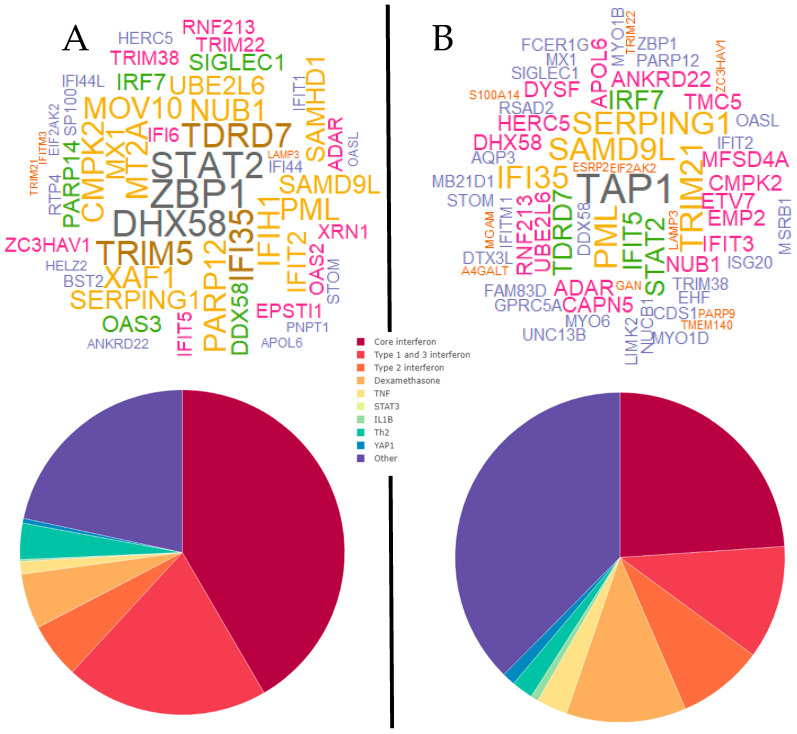
Word cloud plots of the top 20 hub genes for each individual at the acute (Day 1–3) visit (**top**) and averaged network pathway proportions (**bottom**) for clusters A and B. Word color and size indicates frequency.

**Figure 11 jpm-11-01293-f011:**
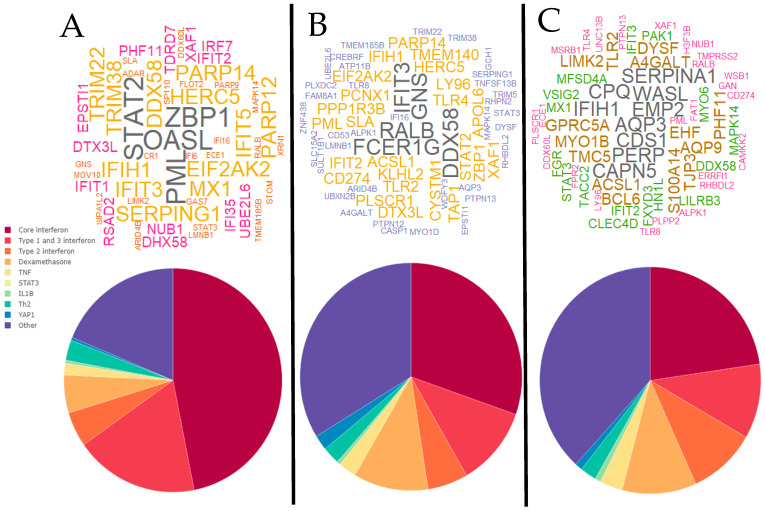
Word cloud plots of the top 20 hub genes for each individual at the late (Day 4–6) visit (**top**) and averaged network pathway proportions (**bottom**) for clusters A, B, and C. Word color and size indicates frequency.

**Table 1 jpm-11-01293-t001:** Characteristics of the study population used for microarray analysis of gene expression.

	RV-Negative Controls	RV-Negative Convalescence	RV-Negative Cases	RV-A Cases	RV-C Cases
*n*	21	14	19	11	26
Age in years, mean (SD)	6.62 (4.50)	4.18 (1.67)	3.47 (3.26) ^#^	6.42 (3.30) *	4.11 (3.09) ^#^
Males, *n* (%)	8 (38)	7 (50)	10 (53)	6 (55)	14 (54)
Overall atopy, *n* (%)	10/17 (59)	7/11 (64)	8 (42)	10 (91) **	16 (62)
Atopy to aeroallergens only, *n* (%)	10/17 (59)	7/11 (64)	8 (42)	9 (82) *	14 (54)
Children with virus detected, *n* (%)	8 (38)	3 (21)	12 (63)	11 (100) *^###^^†††^	26 (100) ***^###^^†††^
Time between first symptoms and hospital presentation in days, mean (SD)	NA	NA	5.21 (4.22)	5.00 (4.56)	3.62 (2.50)
Children diagnosed with acute asthma, bronchiolitis or wheeze, *n* (%)	NA	NA	17 (89)	11 (100)	26 (100)
Exacerbation severity Z score, mean (SD)	NA	NA	0.56 (0.79)	0.44 (0.76)	0.43 (0.88)
Acute use of corticosteroid, *n* (%)	NA	NA	11/14 (79)	8/8 (100)	15/17 (88)

* *p* < 0.05 compared with RV-negative cases; ** *p* < 0.01 compared with RV-negative cases; *** *p* < 0.001 compared with RV-negative cases; ^#^ *p* < 0.05 compared with RV-negative controls; ^###^ *p* < 0.001 compared with RV-negative controls; ^†††^ *p* < 0.001 compared with RV-negative convalescence; NA = not assessed.

**Table 2 jpm-11-01293-t002:** Clinical characteristics of Clusters A, B, C, and D.

	Cluster A	Cluster B	Cluster C	Cluster D
*n*	19	16	10	11
Age in years, mean (SD)	4.10 (2.67)	4.21 (2.67)	5.42 (5.24)	3.99 (3.24)
Males, *n* (%)	12 (63)	10 (62)	3 (30)	5 (45)
Overall atopy, *n* (%)	14 (74)	11 (69)	5 (50)	4 (36) *
Atopy to aeroallergens only, *n* (%)	13 (68)	11 (69)	4 (40)	3 (27) *^†^
Children with virus detected, *n* (%)	18 (95)	12 (75)	8 (80)	11 (100)
RV species, RV-A *n* (%), RV-C *n* (%), RV-negative *n* (%)	3 (16), 9 (47), 7 (37)	3 (19), 7 (44), 6 (37)	3 (30), 3 (30), 4 (40)	2 (18), 7 (64), 2 (18)
Time between first symptoms and hospital presentation in days, mean (SD)	3.63 (3.30)	5.50 (4.05)	5.20 (4.21)	3.55 (2.66)
Children diagnosed with acute asthma, bronchiolitis or wheeze, *n* (%)	18 (95)	16 (100)	9 (90)	11 (100)
Exacerbation severity Z score, mean (SD)	0.55 (0.81)	0.37 (0.92)	0.57 (0.73)	0.42 (0.84)
Acute use of corticosteroid, *n* (%)	9/11 (82)	9/9 (100)	8 (80)	8/9 (89)

* *p* < 0.05 compared with Cluster A; ^†^ *p* < 0.05 compared with Cluster B.

## Data Availability

The data presented in this study are openly available in Gene Expression Omnibus, reference numbers GSE103166 and GSE115824.

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
