# Peer review of "Personal Network Inference Unveils Heterogeneous Immune Response Patterns to Viral Infection in Children with Acute Wheezing"

_jpm, 2021, doi:10.3390/jpm11121293_

Round 1
Reviewer 1 Report
In this manuscript, Dr. Coleman et al. applied an innovative approach to a dataset of gene expression data on children with asthma to identify different immune response patterns related to molecular phenotypes. The manuscript is very well written, the statistical analyses are sound, and the results are of interest in the field of personalized medicine.
Major comments
- Lines 78-80: controls were siblings or friends of the patients included or individuals from the community. Therefore, there is an admixture of family-related and unrelated individuals. Please, indicate the proportion of related individuals and discuss how including related individuals could affect the results.
- Line 90: there is no specification about any specific procedure to preserve RNA in the samples analyzed. Please, add more details about this.
- Was the microarray data validated by assessing gene expression of selected genes by real-time PCR? Please, discuss.
- Figure 1: figure legend should be more explanatory, and the quality of the figure need to be improved. The names of the probes IDs in the figure are not legible. It would be good to add the names of the genes where the probes are located next to their IDs.
- Figure 4: it is very hard to distinguish the genes included in the network. Consider moving this figure to the supplement, providing each panel at a bigger size or deleting it from the manuscript.
- Line 669-671: one of the limitations of this study is the analysis of a heterogeneous tissue. In line 109, the inclusion of sva components as covariates in the models is mentioned. Could the results be confounded by differences in cell composition? Do sva components capture cell heterogeneity within the nasal samples?
- Supplementary figures were not available for review.
Minor comments
- Lines 27-28: based on the results of the study, the statement “These findings demonstrate the applicability of personal network inference for biomarker discovery and drug development” is too optimistic. This reviewer suggests replacing “drug development” with “therapeutical target identification”.
- Line 45: the name of the CDHR3 gene should be written in italics.
- Line 63: although it is easy to infer that “IRF7hi” and “IRF7lo” refer to IRF7 high and IRF7 low, it would be better to spell out the names of the two groups.
- Lines 345-348: please, remove these lines.
- Figure 5: what does the NA category represent? Does it refer to the control group?
- Table S1: the names of the genes in the second column should be written in italics.
Reviewer 2 Report
This manuscript by Laura A Coleman et al. studied the gene signatures in virus-induced exacerbations through personal network inference. The results are interesting, but some parts still need to be improved.
- Although you showed detailed information about different groups in Table 1, but it is better to explain what are the differences among them, and why?
- In Table 1, what statistical analysis you performed here?
- You should add some background about personal network inference in introduction.
- Line 182, what is T2 inflammation?
- In Figure 1, The names of groups were different with Table 1, I suggest you used the same name.
- in the beginning, you chose 5 different groups, and after Figure 2, you just talked about 4 clusters. what is the relationship between them? I think your point is about 4 clusters, if so, it is necessary to show 5 different groups?
Round 2
Reviewer 2 Report
The author already addressed all of my comments in current manuscript.